# A Field Data Acquisition Method and Tools for Hazard Evaluation of Earthquake-Induced Landslides with Open Source Mobile GIS

**Mauro De Donatis ***[iD], **Giulio F. Pappafico** and **Roberto W. Romeo**

Department of Pure and Applied Sciences, University of Urbino 'Carlo Bo', Urbino 61029, Italy; giulio.pappafico@uniurb.it (G.F.P.); roberto.romeo@uniurb.it (R.W.R.)

**\*** Correspondence: mauro.dedonatis@uniurb.it; Tel.: +39-0722-304-295

**Abstract:** The PARSIFAL (Probabilistic Approach to pRovide Scenarios of earthquake Induced slope FAiLures) method was applied to the survey of post-earthquake landslides in central Italy for seismic microzonation purposes. In order to optimize time and resources, while also reducing errors, the paper-based method of survey data sheets was translated into digital formats using such instruments as Tablet PCs, GPS and open source software (QGIS). To the base mapping consisting of Technical Regional Map (Carta Tecnica Regionale—CTRs) at the scale of 1:10,000, layers were added with such sensitive information as the Inventory of Landslide Phenomena in Italy (Inventario dei Fenomeni Franosi in Italia—IFFI), for example. A database was designed and implemented in the SQLite/SpatiaLite Relational DataBase Management System (RDBMS) to store data related to such elements as landslides, rock masses, discontinuities and covers (as provided by PARSIFAL). To facilitate capture of the datum on the ground, data entry forms were created with Qt Designer. In addition to this, the employment of some QGIS plug-ins, developed for digital surveying and enabling of quick annotations on the map and the import of images from external cameras, was found to be of considerable use.

**Keywords:** field mapping; QGIS; NMEA GPS; RDBMS

## 1. Introduction and Aims

Acquiring data in the field is a common practice for geologists. The traditional system, which is still in use, involves data collecting on maps and on paper notebooks, and from this the transition is being made to digital gathering of data and information. When timing, accuracy, and group work are of the essence, digital methods allow quick, shared, and accurate results to be achieved [1].

For the seismic microzonation work of the area between Arquata and Pescara del Tronto, carried out following the seismic event of 24 August 2016 in central Italy, geological/technical surveys were completed on the earthquake-induced landslides.

For this purpose, the PARSIFAL (Probabilistic Approach to pRovide Scenarios of earthquake Induced slope FaiLures) method [2–4] was used to analyse the data that were initially acquired using paper survey data sheets, and then transferred this information into a GIS database. At any rate, considering that, for a number of years, a variety of systems have been in use for the digital acquisition of data in the field [5–7]), mobile GIS was subsequently adopted in order to be able to digitally survey and archive data and information directly in the field. The digital survey system, which has been in use for years for landslide mapping [8–11], was prepared as needed, providing it with a database and forms to acquire data ready for subsequent processing.

The aim of this paper is to explain a new relational database (SQLite/SpatiaLite) and provide a free field digital tool based on an open source GIS (see Appendix A) for any surveyor working with the PARSIFAL method.

## 2. The PARSIFAL Method

Different methods of hazard assessment for earthquake-induced landslides are proposed in the scientific and technical literature [12–14].

The PARSIFAL method was elaborated by CERI-Sapienza (Centre for Research in Forecasting, Prevention and Control of Geological Risks of "La Sapienza" University) in collaboration with Università di Urbino and ENEA (National Agency for New Technologies, Energy and Sustainable Economic Development). Firstly, it was trialled in certain municipalities of southern Lazio [2]. Then, it was applied in the territory of the municipality of Accumoli (Central Italy) after the series of quakes of 2016–17 [3] and in the basin of Alcoy (Alicante, South Spain) [4].

The PARSIFAL method has the following main characteristics:

- it is set up to assess the hazard posed by first- and second-generation landslides (which is to say landslides that are newly activated, typically coseismic, and reactivations of pre-existing landslides, and as such ones that can already be censused prior to a seismic event);
- it performs analyses differentiated by the landslide mechanism (e.g., sliding and overturning of blocks of stone, sliding in the earth, etc.);
- it plots summary mapping with indications on the probability of going beyond the thresholds of the coseismic shift if they are exceeded, or on the margins of safety in inappreciable seismic conditions due to shifting, or purely roto-translational kinematics (overturning);
- it permits a probabilistic analysis, weighted for each map unit, of the expected effects in terms of the landslide mechanism and its intensity.

In particular, the elements required for assessment using PARSIFAL are the data on: (i) "rock masses" of the geological substratum present in the survey area; (ii) "covers," for each unit of Quaternary cover outcrop along the slopes; and (iii) "landslides," both newly activated and pre-existing.

## 3. The System for Gathering Data on the Ground for PARSIFAL

To guarantee orderly and complete acquisition of the geological/technical data necessary for applying PARSIFAL in the areas affected by the seismic microzonation studies done after the Amatrice quake, paper modules were prepared for the gathering of data by the surveyors (Figure 1). The modules had the purpose of guaranteeing the systematic, univocal gathering of the necessary information, and of easing the acquisition work without hindering the priority surveying [15]. The modules refer to a single observation/measurement station. To aid the surveyor, tables were also provided for definitions and classes from which to obtain the codes of the specific geological/technical data to be entered into the modules (Figure 2).

Specifically, there were three paper modules. Each of them was designed to gather the data on the three types of subjects observed by the surveyors for the purposes of applying PARSIFAL: rock masses, covers, and landslides.

The study of the modules directly provided the properties characterizing the database implemented in this work, as there is no other documentation beyond these modules.

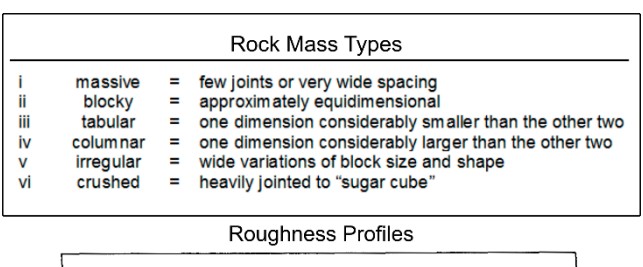

**Figure 1.** Example of the form (for rock masses) for data collecting used in the field survey (modified from the original form in Italian).

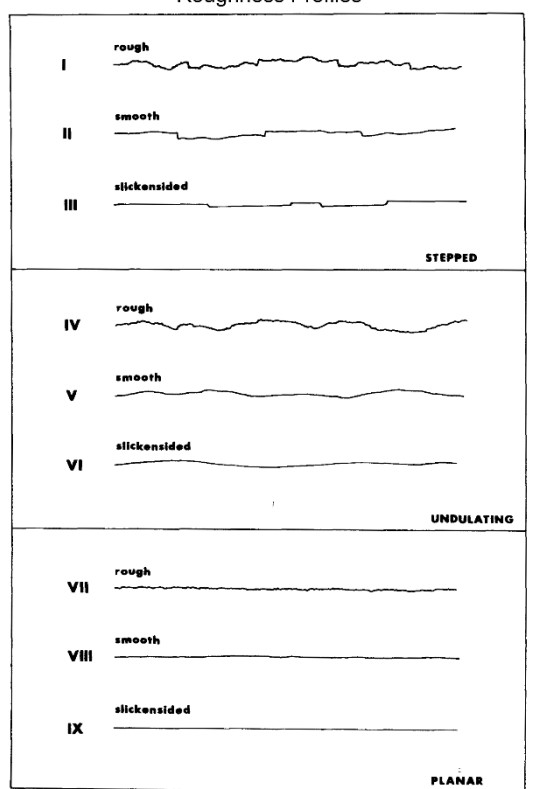

**Figure 2.** Example of tables for geotechnical data codes (adapted from ISRM [16]).

## 4. Hardware

The fundamental instruments used in this work were:

- A Microsoft Surface Pro 3 Tablet PC (Intel Core i5, Processor 256 Gb of SSD and 8Gb of RAM) with Windows 10 operating system, which allowed for very rapid management of the thematic mapping and of the georeferenced raster aerial images;
- A 51-channel Bluetooth GPS receiver (NMEA protocol, WAAS EGNOS correction), since the tablet that was used has no internal GPS receiver, and because the one employed guarantees greater precision.

The following accessories were also used:

- a dedicated digital pen with Bluetooth link, to be used in place of a mouse, and to draw and write notes using dedicated software tools;
- a dedicated cover in plastic and rubber with ergonomic support at the wrist for the tablet's transportability and protection;
- several different models of digital cameras.

These instruments have the advantages of lightness and manageability, including in "uncomfortable" conditions like those operators in the field often have to deal with, and as such they are easy to use. The costs which, while being higher than for a laptop with similar technical characteristics, have diminished drastically over the years, permitting their greater availability among geologists.

Lastly, there is no denying that the use of the pen on the screen, in addition to greatly facilitating operations that would normally have to be done with a mouse, makes it possible to maintain the "feel" of the traditional, "pen-on-map" surveying method.

Of great importance, however, is the availability of software packages allowing the best achievable exploitation of the possibilities the hardware has to offer.

## 5. Software

The software used was QGIS (version 2.18), which has been employed for years in field research. The choice of this GIS software is linked to its simple use and to its spread, as well as to the Open Source approach that allows the customization and creation of new tools suited to the requirements of the work in the field.

In particular, some plug-ins conceived in the laboratory were developed, thanks to the efforts of some students in their thesis work, and those of outside collaborators [17].

These include BeePen, for free-handed drawing and writing, BeeGPS, to manage GPS data, and BeePic, for the georeferencing of the photos.

BeePen creates particular layers of annotation and permits writing by digital pen directly on the map, choosing colours, transparencies, and line thickness, while also inserting notes. Using this tool, the surveyor can use the tablet's screen like a map on which to annotate any element, as can be done on paper maps.

BeeGPS adds functionalities to the GPS module already existing in QGIS, making it possible to enter values that personalize the acquisition of GPS data by choosing an acquisition interval (in seconds). This option is useful when taking points in movement on different modes of transport (on foot, by bike, by car, etc.). The distance threshold, within which the software does not record data, is helpful when the operator stops to make observations or acquire data, thereby eliminating the useless and annoying cloud of points typical of this kind of GPS [18].

BeePic makes it possible to georeference photographs taken using any camera or smartphone, including those not equipped with GPS. The acquisition of the GPS of the mobile GIS, and the EXIF format of the images—containing both sets of temporal information—allow the photos to be synchronized and therefore geolocalised.

Moreover, using QT Designer installed along with QGIS, the data acquisition forms—the central object of this work—can be created and customized.

## 6. Database

An SQLite/SpatiaLite geographic database was implemented in accordance with the scheme shown in Figure 3, in which the tables *lSlide*, *rMass*, *cover* and *disc* contain the attributes corresponding to the fields present in the paper modules for acquiring the geological/technical data needed to apply PARSIFAL.

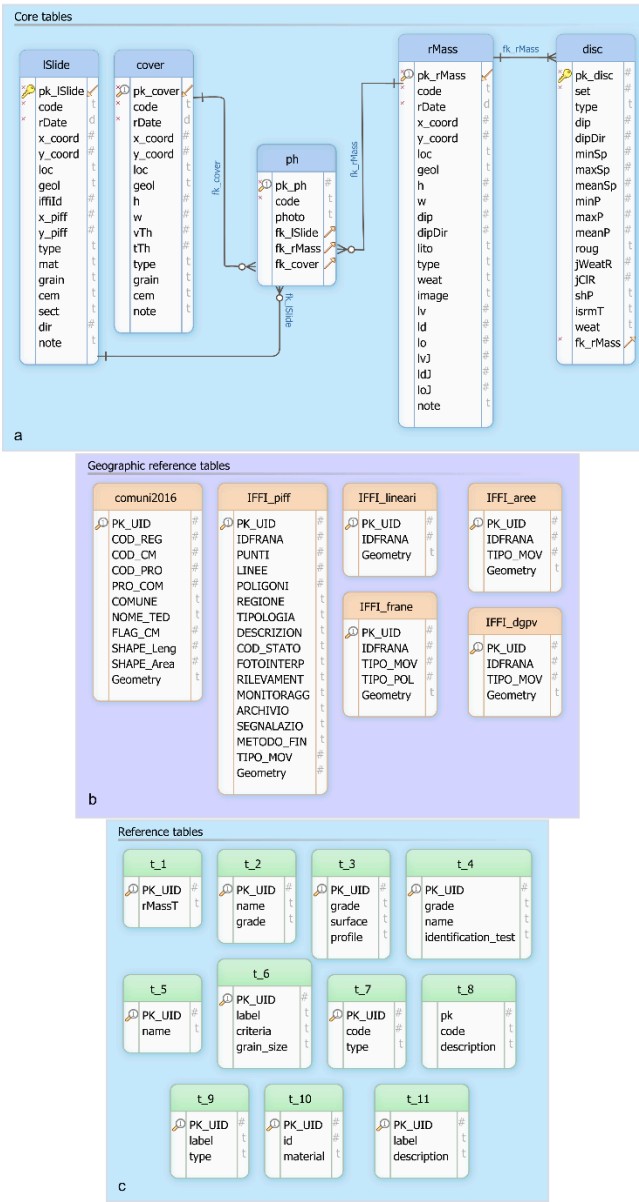

**Figure 3.** *Parsifal* database schema: (**a**) core tables; (**b**) reference geographic tables; (**c**) reference tables for geotechnical data coding (for all tables attributes and codes see the HTML file mentioned in Appendix A and linked here http://www.geologiapplicata.uniurb.it/download/parsifal.html).

The database, called *parsifal*, contains five main tables and 17 reference tables (Figure 3). The structure of the five main tables (Figure 3a) reflects the organization of the PARSIFAL method's modules and meets all the prerogatives described in the note by Della Seta et al. [15] and Martino et al. [4]. Three of the five tables store the information regarding the landslide bodies, already censused or newly

identified, the fractured rock masses, and the cover terrain. The three tables are respectively named "lSlide," "rMass," and "cover" (Figure 3a), and also contain the type of *geometry* datum for representing the spatial-point elements.

Regarding the rock masses, in particular, the PARSIFAL method also calls for entering data regarding the discontinuity systems that relate to the mass, for which an additional table (*disc*) is provided. This table describes all the information regarding the geometry and characteristics of the discontinuity (Figure 3a). The two tables, *rMass* and *disc*, are then associated by means of a "one-to-many" relationship.

The photographic documentation is stored in the *ph* table associated with the three tables—*lSlide*, *rMass*, and *cover*—by means of "one-to-many" relationships (Figure 3a). Each individual landslide, rock mass, or cover terrain can be documented by one or more photographic images.

The database also includes six geographic reference tables (Figure 3b) which serve to recover the data regarding the landslides in the inventory of landslide phenomena in Italy (Inventario dei Fenomeni Franosi d'Italia—IFFI [19]), as they are pre-existing landslides. Moreover, it links the recorded information to its territory, by means of the municipality's ISTAT (Italian National Institute of Statistics) code. Eleven other tables code the specific geological/technical data required by the PARSIFAL method (Figure 3c), such as the type of rock mass, its degree of alteration, the roughness of the profile of the rock surface, the degree of alteration of the rocky material, and the grain size classification of the landslide material.

The attributes of the database's tables are linked to the fields of a series of forms made using QT Designer. The forms are designed for use within the QGIS graphic interface; they, in turn, constitute the graphic interface with the PARSIFAL geographic database, which allows the surveyor to quickly record the data whose proper entry and integrity the database is ensured.

The SQL source code for creating the database, and the corresponding documentation, can be downloaded from the links reported in Appendix A.

## 7. Input Forms

Three forms were configured for the purpose of easing the acquisition of data on the ground and storing it directly in the "parsifal" database, and they correspond to the main elements being surveyed: landslides, rock masses, and covers. Each of the three forms is a tabbed form composed of multi-pages, accessible through the respective tabs (Figure 4a,c,d).

The first form page (Figure 4a), common to the three types of elements observed, allows the common data (measuring station, surveyor, and date) to be entered. The station coordinates ("GPS Coordinates") are automatically recorded thanks to the action of a trigger, which is executed at the moment of the digitization of the point representing the station (see the SQL code mentioned in Appendix A, from line 111 to line 117, for the "Landslide" element; from line 283 to line 289 for the "Mass" element; and from line 413 to line 419 for the "Cover" element).

The "Landslide Code," "Rock Mass Code," and "Cover Code" fields, in the respective "Measuring station" form pages, are also compiled by the action of the respective triggers performed at the moment the data are saved (from line 82 to line 106 for the "Landslide Code"; from line 254 to line 278 for the "Rock Mass Code" and from line 384 to line 408 for the "Cover Code" in the SQL code mentioned in Appendix A).

The second form page, named "Landslide" in the case of landslides, "Rock Mass" in the case of rock masses, and "Cover" in the case of covers, contains the attributes whose values must be recorded for the application of the Parsifal method, and differs depending on the type of element observed.

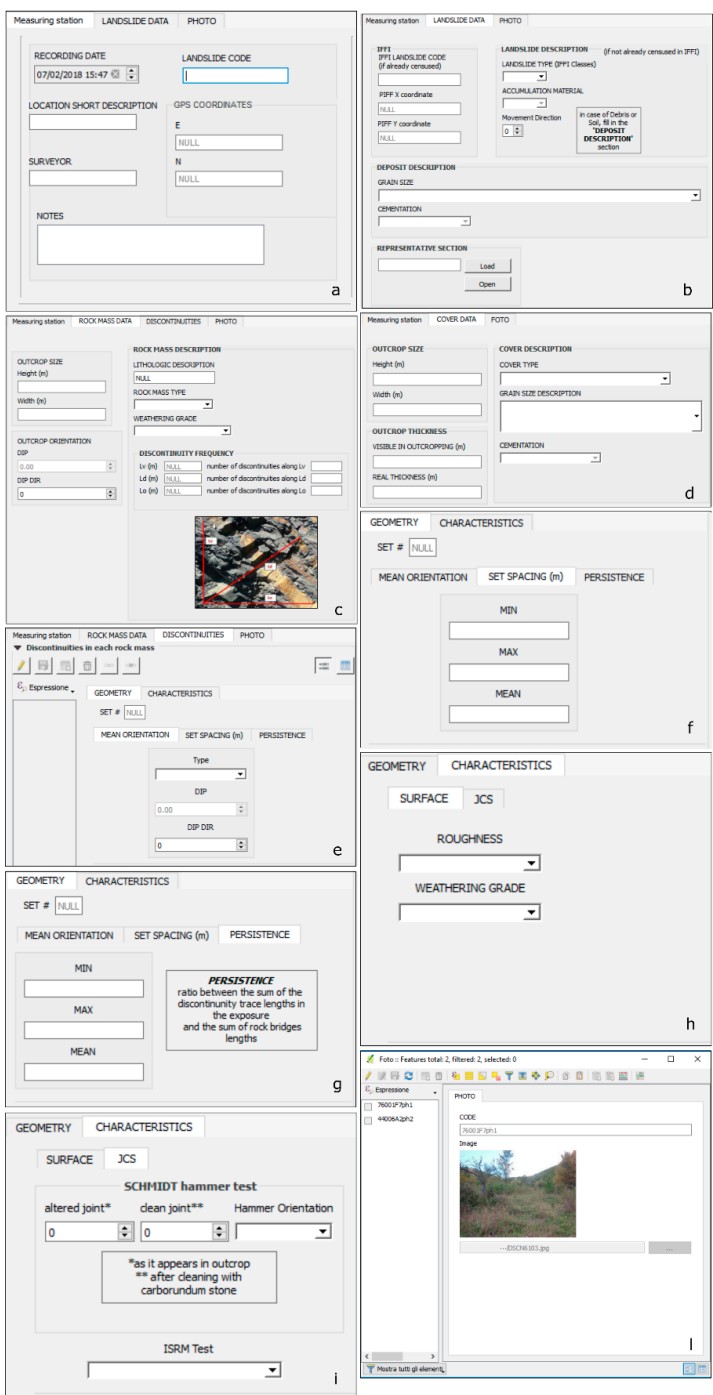

**Figure 4.** Data acquisition forms: (**a**) Landslide data recording form where the "Landslide Code" and "GPS Coordinates" fields are automatically filled; (**b**) "Landslide" tab attributes; (**c**) Attributes of the "Rock Mass" tab; (**d**) Attributes of the "Cover" tab for recording cover Quaternary deposits data; (**e**) "Discontinuities" tab showing the two main form pages "Geometry" and "Characteristics"; (**f**) Fields of "Set Spacing (m)" tab where minimum, maximum and mean spacing values of a single set of discontinuities can be recorded; (**g**) Fields of "Persistence" tab where minimum, maximum, and mean persistence values of a single set of discontinuities can be recorded; (**h**) "Characteristics" tab showing the form pages "JCS" and "Surface" where the "Roughness" and "Weathering Grade" fields are included; (**i**) "JCS" form page showing the fields to be populating with the joint wall compressive strength test results; (**l**) "Photo" tab including "Photo Code" showing the image preview and file path.

1.　In the case of landslides, the attributes are (Figure 4b):

- "IFFI Landslide Code", to be compiled in the event of a landslide already present in the inventory of landslide phenomena in Italy (IFFI). The code's value may be obtained by querying the PIFF point (*Punto Identificativo del Fenomeno Franoso*—"landslide event identification point") from the map and copying the value contained in the "IDFRANA" field, the univocal identifier of the landslides on the national inventory.
- "PIFF X coordinate" and "PIFF Y coordinate" of the IFFI's landslide event identification point (PIFF): the fields of the two attributes are automatically populated once the "IFFI Landslide Code" is entered and the data are saved, through execution of the trigger as per line 124 to line 147 of the SQL code mentioned in Appendix A.
- "Landslide Type", "Accumulation Material", "Grain Size", and "Cementation" are attributes whose values may be selected from the respective drop-down lists present in the form. The fields are linked to the attributes of the database's reference tables containing the codes of the geological/technical data required by the Parsifal method (Figure 5a).

2.　In the case of rock masses, the fields to be compiled consider the following attributes (Figure 4c):

- "Height" and "Width" to enter the outcrop's dimensions.
- "Outcrop Orientation" to record bedding measurements ("DIP" and "DIP DIR").
- Three fields which, together, contribute to the description of the rock mass: "Lithologic Description", "Rock Mass Type", and "Weathering Grade", whose values may be selected from the respective drop-down lists present in the form. In this case the fields are linked to the attributes of the database's reference tables containing the codes of the geological/technical data required by the Parsifal method (Figure 5b).
- "Discontinuity Frequency" is a characteristic defined by the set of six attributes represented by the "number of discontinuities along 'Lv' | 'Ld' | 'Lo'"—respectively, for the directions Lv, Ld, Lo, their lengths expressed in metres are entered.

3.　In the case of covers, the fields to be filled in consider the following attributes (Figure 4d):

- "Height" and "Width" to enter the dimensions related to the outcrop.
- "Outcrop Thickness" to be evaluated to the visible cover in outcropping, and "Real Thickness (m)" when this may be measured.
- "Cover Type", "Grain Size Description", and "Cementation", whose values may be selected from the respective drop-down lists present in the form (Figure 4d). In this case the fields are linked to the attributes of the database's reference tables containing the codes of the geological/technical data required by the PARSIFAL method.

The "Discontinuities" form page (Figure 4e) is used to record a series of parameters necessary to characterize the discontinuity sets recognized in the mass being observed. The discontinuity sets may be one or more than one in number; the "SET #" field visible under the "Geometry" tag shows the number of discontinuity sets to which the parameters refer.

The data on the discontinuities are stored in the "parsifal" database's "disc" table. This table is associated with the "rMass" table by means of an external key in accordance with a one-to-many relationship, and therefore each mass is associated with one or more discontinuity sets.

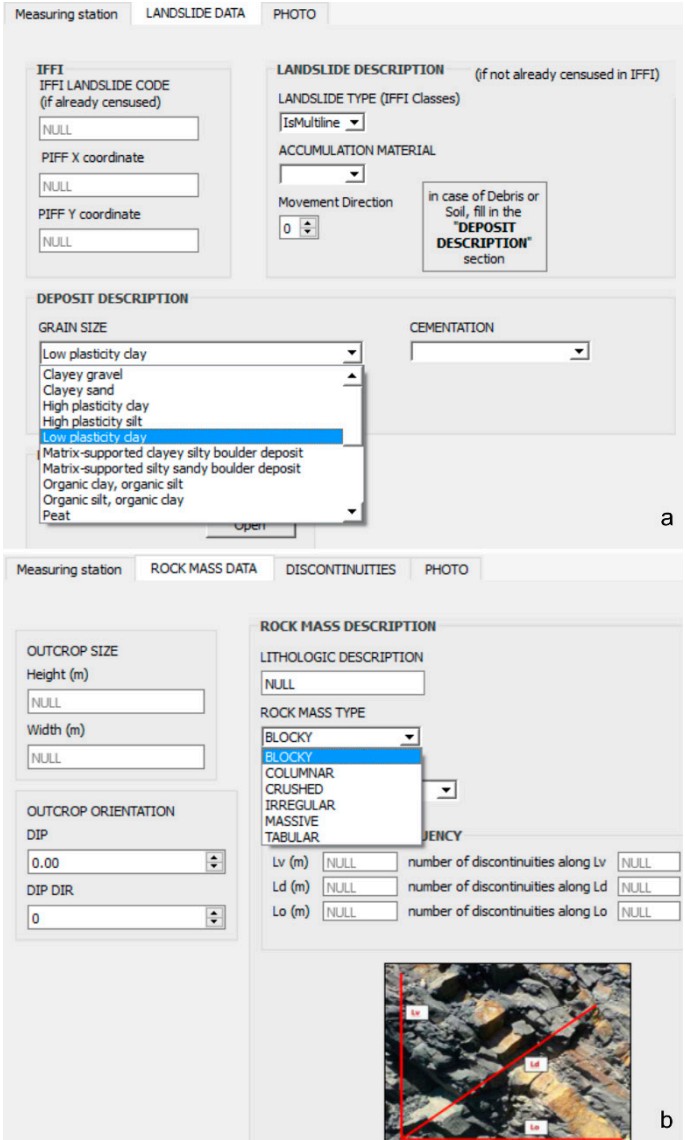

**Figure 5.** Input forms with combo boxes containing a drop-down list of the geotechnical data values required by the Parsifal method: (**a**) example of "Grain Size" attributes; (**b**) example of "Rock Mass Type" attributes.

For each set, it is possible to record a set of parameters distributed in two additional form pages of the "Discontinuity" form: "Geometry" and "Characteristics" (Figure 4e).

a.    The "Geometry" form has three additional form pages:

1.    "Mean Orientation": the fields to be compiled consider the following attributes:

- "Type": values may be selected from a drop-down list of discontinuity types.
- "DIP": inclination of the discontinuity surface.
- "DIP DIR": direction of immersion of the discontinuity.

2.    "Set Spacing (m)": contains the fields related to the attributes (Figure 4f):

- "MIN": minimum spacing of the individual discontinuity set.
- "MAX": maximum spacing of the individual discontinuity set.
- "MEAN": average value of the spacing of the individual discontinuity set.

3. "Persistence": presents the following fields to be compiled (Figure 4g):

- "MIN": minimum value of the persistence of the individual discontinuity set.
- "MAX": maximum value of the persistence of the individual discontinuity set.
- "MEAN:" average value of the persistence of the individual discontinuity set.

b. The "Characteristics" form (Figure 4h) has two additional pages:

1. "Surface": the fields to be compiled, which consider the following attributes:

- "Roughness": descriptive of the ISRM [16] standards, and may be selected from a drop-down list.
- "Weathering Grade", whose values express the degree of alteration in accordance with ISRM [16] standards, and may be selected from a drop-down list.

2. "JCS" (Joint wall Compressive Strength), which marks the tab, is the compressive strength of the joints' walls. It is defined by the values of the characteristic results of the "Schmidt hammer test" and of the "Manual index test" (ISRM Test) to be entered into the following fields (Figure 4i):

- "altered joint".
- "clean joint".
- "Hammer Orientation": the description may be selected from a drop-down list.
- "ISRM Test": this is intended as the result of the "Manual index test", whose value may be selected from a drop-down list.

The third form page ("Photo") is common to the three elements that may be observed (landslides, rock masses, and covers), and is used to enter the information regarding the images depicting their characteristics (Figure 4l). The data related to the images are stored in the "parsifal" database's "ph" table. This table is associated with the "lSlide", "rMass", and "cover" tables, using the external keys in accordance with one-to-many relationships, for which each landslide, mass, or cover is associated with one or more images. The form page has the following fields:

- The "Photo Code" is compiled by the action of a series of triggers (from line 491 to line 537 of the SQL code mentioned in Appendix A) performed at the moment the data are saved.

The photo's code is composed of the code previously assigned by the system to the element observed on the ground to which the image refers (landslide, rock masses, or cover), by the initial ("L" in the case of landslide, "R" in the case of rock mass, and "C" in the case of cover), and by the primary serial key of the recorded image.

- "Photo": the name of the "ph" table's field present in the database, in which the image's path and file name are stored. To load an image's path and file name from the folder where it was saved and store it in the database, the QGIS "Photo" widget is used, which also allows the image to be previewed in an area of the form page (Figure 4l).

## 8. Other Data and Information

In addition to acquiring data "structured" in tabular form using the survey sheets system illustrated above, mobile GIS, customized with certain plug-ins and other accessory programs, made it possible to capture and manage information that was useful for the final processing. In particular, the BeePen module allowed the digital pen to draw certain notes quickly and directly on the monitor map (Figure 6). In fact, speed is an important requirement in digital fieldwork. In this case, for example, some quick corrections were made to the landslide and outcrop limits that were available from the

IFFI mapping (Landslides Catalogue from the "Italian National Geoportal") or new acquisitions of landslides not surveyed earlier were made, perhaps while entering some indications and comments. These lines were then digitized in a new layer.

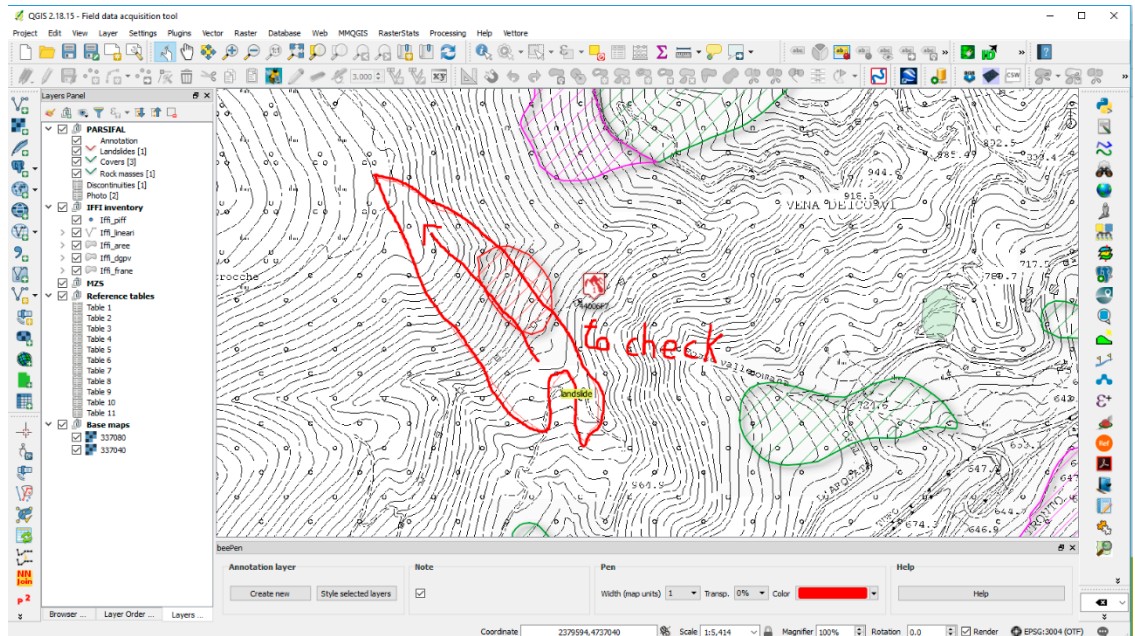

**Figure 6.** QGIS window with layers of the PARSIFAL project. The BeePen plug-in (at the bottom) allows sketches and annotations on the map, as shown.

BeePic, on the other hand, made it possible to geolocalise the photos taken with a number of devices, and also by different operators. This method is the result of extreme utility in positioning the picture-taking points on subsequent days in the laboratory, thus making the work in the field quicker and more efficient.

## 9. Comparative Experiences on the Field and in the Laboratory

The work of data gathering in the field was done for some days by two surveyors. The first used paper sheets and maps, while the second used the digital tools described above.

At the start of the survey, the "analog" operator gathered data using paper sheets, maps, a field notebook, and a digital camera.

The collected photos, sheets, and information reported on the map were then transcribed and digitized, georeferencing them to a GIS project after it was back in the laboratory.

The "digital" operator carried out the survey using the same procedures as the first operator, but accelerating and often improving the positioning thanks to the GPS. The greater difference, at any rate, was the entry of the data within the GIS project directly in the field. Some photos were entered into the sheets directly and others at a later time, both during some phases of interruption of the survey and in the laboratory, using BeePic. In addition to reporting the points of the stations, it was possible to draw in the same manner as drawing on paper—which is to say using a pen on the map—the indications of landslides not reported by the official inventories (some of very recent activation), which were then redrawn on a special layer in the laboratory.

The first, "analog" case required a longer working time. Moreover, in some cases, it was modified in subsequent checks, where errors, doubts, or inconsistencies were highlighted.

In the digital survey, subsequent laboratory work was also needed to better arrange or complete tables and photo insertions. However, in addition to taking place in a decisively quicker timeframe than in the first case, doubts and errors were promptly resolved by the sketches or annotations outlined

directly on the map. It was then possible to maintain in the design both the layers used for the survey and the subsequent ones of laboratory synthesis of the mapping product and final databases.

## 10. Conclusions

In this experience, in nearly emergency conditions, the digital surveying method turned out to be a considerable aid both in the acquisition phase and in the data processing phase.

In particular, the method's strong points may be defined as follows:

1. Surveying efficiency: the times both of the survey on the ground and those following processing and synthesis were reduced.
2. Precision in positioning: the GPS that made it possible to follow the movements and the positioning of the station points in many cases were more accurate, particularly when far from certain map references (e.g., climbing up a ditch in a wooded area).
3. Elimination of errors due to subsequent transcriptions/digitizations: this error is frequent during the transfer of data from a paper map to a digital one [20], in addition to the transcription of data from paper sheets to database tables. In this case, the intervention was completed with changes after entry, while this further checking work was not necessary for the data digitally captured in the field.
4. Maintenance of the traditional "pen-on-map" surveying system: in some way, the traditional surveying system was maintained thanks to the use of tools that display the mapping and permit writing and drawing as can be done on paper.
5. Ability to acquire "non-structured" information of importance for subsequent interpretations: the information reported through annotations with signs and drawings on the map was of use for processing with greater precision in the laboratory (when it was a matter, for example, of redrawing the landslide bodies), and also for recalling certain elements of use for processing data.
6. Simplification of group work among surveyors: although in this project the two surveyors actually used two different systems (analog and digital), the simplification in gathering data digitally is clear, thereby in some cases reducing surveyor subjectivity.

**Author Contributions:** Conceptualization, M.D.D., G.F.P. and R.W.R.; Funding acquisition, M.D.D.; Methodology, M.D.D.; Project administration, M.D.D.; Software, G.F.P.; Supervision, M.D.D.; Validation, M.D.D. and R.W.R.; Writing—original draft, M.D.D., G.F.P. and R.W.R.; Writing—review & editing, M.D.D., G.F.P. and R.W.R.

**Funding:** The founding sponsors had no role in the design of the study; in the collection, analyses, or interpretation of data; in the writing of the manuscript, and in the decision to publish the results.

**Acknowledgments:** This work is part of the project "Identification and characterization of the system of seismogenic faults in the earthquake of Cagli in 1781" of "Research enhancing program 2017" funded by DISPEA (Department of Pure and Applied Sciences) of Urbino University.

**Conflicts of Interest:** The authors declare no conflicts of interest.

## Appendix A

The RDBMS SQLite/SpatiaLite database is available as an SQL file at this web page: https://listauniurb.jimdo.com/parsifal/, where an HTML file (parsifal.html) is also available for textual and graphic documentation of the database tables and fields.

A QGIS project ready for field survey with data entry forms and IFFI (Inventory of Landslide Phenomena in Italy) database can be downloaded from the same web page.

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
