# Peer review of "A Field Data Acquisition Method and Tools for Hazard Evaluation of Earthquake-Induced Landslides with Open Source Mobile GIS"

_ijgi, doi:10.3390/ijgi8020091_

Round 1

Reviewer 1 Report

The paper fits perfectly with the aim of the special issue, given that it presents a on-field survey digital solution for landslide mapping in the view of the PARSIFAL protocol. The digital survey tools presented seems effective in improving field geological mapping. The main concerns are related to the english language that should be a little improved (some time the flow of the discussion is difficult to follow) and to the final conclusions, where some information on data redundancy (i.e., which kind of solution are used for data redundancy, e.g., cloud uploading?) and on possible future extensions. 

Author Response

Reviewer's comments and authors reply

The main concerns are related to the english language that should be a little improved (some time the flow of the discussion is difficult to follow) and to the final conclusions, where some information on data redundancy (i.e., which kind of solution are used for data redundancy, e.g., cloud uploading?) and on possible future extensions.

We sent our text at language editing service to be improved.

Proper constraints and the application of the main database normalization rules help to avoid data redundancy both when storing data in the field, and when uploading data on a centralized database during the following laboratory processing. The choice of the local database solution instead of the client-server one was preferred due to the not always available network in the field work. Moreover our mobile tool was ideated for capturing data only. Being an open source solution, any person interested on solve any kind of eventual problem can work on that and offer other implementation.

We correct the text as suggested in the peer-review-3494263.v1.pdf

Reviewer 2 Report

The article presented is up-to-date and presents something innovative.

Did the authors of the article have contributions to the data acquisition and processing, or were they helped by other colleagues?

The Parsifal Program, reports, and what are they?

Are they text and graphics?

Did the measurements have been made with geodesic class GPS?

Is the database public and can be accessed by any person or managed by an institute or university?

Have you made comparisons with similar products in Italy and the world?

Author Response

Reviewer's questions and authors reply

Did the authors of the article have contributions to the data acquisition and processing, or were they helped by other colleagues?

Only the authors operated on data acquisition and processing. Other people worked on the same area survey using the common paper based method.

The Parsifal Program, reports, and what are they?

What is Parsifal is mentioned in the text, if the question ask for it. I do not understand the question.

Are they text and graphics?

Sorry, I do not understand the question.

Did the measurements have been made with geodesic class GPS?

As mentioned in the 4th chapter “Hardware”, we use a cartographic GPS with NMEA protocol. The accuracy of this kind of receiver is suitable for the base map we used for surveying at the scale of 1:10,000.

Is the database public and can be accessed by any person or managed by an institute or university?

The SQLite/SpatiaLite database is an open-source local (not client/server) DBMS implemented in a cross-platform single monolithic file. That does not require installation nor configuration and that can be freely downloaded and used either independently from specific software or connected to QGIS project, as mentioned in Appendix A of the manuscript. This kind of implementation was preferred because it allows the easy, local and multiplatform use of the database system by field surveyors.

Have you made comparisons with similar products in Italy and the world?

We do not know similar solutions for mapping earthquake induced landslides, even if for almost 15 years we are working on mobile GIS survey solutions. Some previous experience made on landslide mapping (see Gallerini & De Donatis, 2007 and Gallerini et al., 2005) are cited in the text.

Reviewer 3 Report

This paper presented an open source mobile GIS based field data acquisition method and related tools for landslide hazard evaluation. The proposed method followed the PARSIFAL method. In the paper, the authors described the method and the developed tools in detail from different aspects. The structure of the paper is clear, the content is interesting and easy to understand,I only have 1 concern that the authors may need to address in the revision:

while the main purpose of the proposed method is to optimize time and resources and reduce errors, when discussing the use of mobile GIS, it is valuable to add some discussion about collaboration. For example, how multiple surveyor can work together on the same map to add notes and make corrections?

Author Response

Reviewer's questions and authors reply

I only have 1 concern that the authors may need to address in the revision:

while the main purpose of the proposed method is to optimize time and resources and reduce errors, when discussing the use of mobile GIS, it is valuable to add some discussion about collaboration. For example, how multiple surveyor can work together on the same map to add notes and make corrections?

The aim of this work is to provide a tool for any geologist in the field.

Optimization of time and resources is related to the opportunity to have data/information immediately available in digital way. The reduction of errors are related:

- elimination of multiple steps of survey and digitalization.

- accuracy of localization thanks to GPS aid.

Multiple surveyors can work on the same area/map.